# Steady-State Pupil Size Varies with Circadian Phase and Sleep Homeostasis in Healthy Young Men

**Maxime Van Egroo** [†] , **Giulia Gaggioni** [†], **Cristian Cespedes-Ortiz** [†], **Julien Q. M. Ly** and **Gilles Vandewalle** *

GIGA-Cyclotron Research Centre-In Vivo Imaging, University of Liège, Bâtiment B30, Allée du VI Août, 8, 4000 Liège, Belgium; maxime.vanegroo@uliege.be (M.V.E.); giulia.gaggioni@doct.uliege.be (G.G.); c.cespo@gmail.com (C.C.-O.); jly@chuliege.be (J.Q.M.L.)
* Correspondence: gilles.vandewalle@uliege.be; Tel.: +32-4-366-2316
† Joined first authors.

**Abstract:** Pupil size informs about sympathovagal balance as well as cognitive and affective processes, and perception. It is also directly linked to phasic activity of the brainstem locus coeruleus, so that pupil measures have gained recent attention. Steady-state pupil size and its variability have been directly linked to sleep homeostasis and circadian phase, but results have been inconsistent. Here, we report robust changes in steady-state pupil size during 29 h of continuous wakefulness in healthy young men (*N* = 20; 18–30 years old) maintained in dim-light in strictly controlled constant routine conditions. These variations were associated with variations in motivation and sustained attention performance. Pupil size variability did not significantly change during the protocol. Yet, pupil size variability was linearly associated with subjective fatigue, sociability, and anguish. No associations were found between neither steady-state pupil size nor pupil size variability, and objective EEG measure of alertness and subjective sleepiness. Our data support therefore the notion that, compared with its variability, steady-state pupil size is strongly influenced by the concomitant changes in sleep need and circadian phase. In addition, steady-state pupil size appears to be related to motivation and attention, while its variability may be related to separate affective dimensions and subjective fatigue.

**Keywords:** circadian; sleep homeostasis; pupil; motivation; fatigue; sleepiness; attention

## 1. Introduction

The pupil is the window through which the brain captures its visual environment, but the pupil is also a window to the brain. For instance, the pupil dilates to stimuli like fear, surprise, pain, or stress such that emotion processing is a well-known brain function monitored through transient pupil changes [1,2]. Beyond its obvious visual role, pupil diameter is influenced by motivation and cognitive effort, with a larger pupil when performing more difficult tasks [2], or while being more engaged in the task [3]. Transient pupil dilation is also linked to visual awareness and perceptual selection in the viewing of an ambiguous stimulus [4,5].

Pupil size is regulated by the autonomic nervous system and depends on sympathovagal balance. Constriction depends on the parasympathetic nervous system, which directly recruits iris sphincter muscles through the midbrain Edinger-Westphal nuclei [1,6]. Inhibition of parasympathetic tone by the sympathetic system leads to contraction of the dilator pupillae muscle and therefore to pupil dilation [7]. The latter inhibition has received recent scientific attention because it is directly driven by the locus coeruleus (LC). This brainstem structure is part of the ascending and descending activating systems, and is central to sleep and wakefulness regulation, as well as to cognition and anxiety [8]. Direct recording of LC neuron activity in rodents demonstrated that phasic pupil changes, under

constant light condition, are directly related to LC transient bursts [9]. Likewise, human neuroimaging studies have indeed revealed brainstem activity presumably arising from the LC in relation to pupil size variation during an attentional paradigm under constant light condition [10]. Pupil size also varies in response to light stimulation. Pupil Light Reflex (PLR) arises from rod and/or cone light signals that are added to the intrinsic photosensitive responses of retinal ganglion cells (RGC) expressing the blue-sensitive photopigment melanopsin [11]. Intrinsically photosensitive RGC (ipRGCs) project to the olivary pretectal nucleus [11], which in turn projects to Edinger-Westphal nuclei to affect pupil size [1,6]. Following a light stimulation, pupil remains partially contracted for up to tens of minutes (depending on light stimulation characteristics), a phenomenon known as Post-Illumination Pupil Response (PIPR), which is exclusively driven by ipRGC sustained intrinsic response. PLR and PIPR are widespread means to assess the non-visual or non-image-forming functions of light [12–14].

In order to infer about LC activity, emotion processing, motivation, or effort, one needs a full understanding of all the parameters that may affect pupil size and its variability. Several studies reported that steady-state pupil size or pupil size variability under constant light level varies with prior sleep-wake history and could constitute a reliable marker of sleepiness and alertness. The first studies linking sleepiness to pupil changes were carried out in the middle of the 20th century by Lowenstein and Loewenfeld [15–18]. They reported that pupil diameter remained stable in complete darkness but showed increased variability and progressive constriction with increased tiredness, with the smallest pupil immediately prior to sleep. Likewise, they reported that PLR was progressively reduced with increased tiredness with almost no PLR immediately prior to sleep. This absence of response to light was only transient and resumed following an alerting stimulation. They further noticed pupil dilation following an alerting event. These types of results were later confirmed, other pupil measures were considered [1,19–24], and standard tests have been developed, such as the Pupillographic Sleepiness Test (PST) [25]. Pupil size and eyelid closure have been reported to change from well-rested to sleep-deprived daytime conditions [20,23] and hypersomniac patients (e.g., narcoleptic or sleep apnea patients) were reported to show distinct dynamics in pupil size variability during prolonged wakefulness [26,27].

Yet, several studies also failed to find a systematic link between sleepiness and pupil measures. For instance, in 1979, Lavie reported that pupil size and PLR were not systematically related to sleepiness over 10 h of wakefulness, starting around 7–9 AM [24]. In addition, there are indications of changes in pupil size with time-of-day or circadian phase under constant light level [21,24,28,29], i.e., possibly not in relation with sleepiness, although sleepiness is influenced by both sleep homeostasis and circadian phase [30]. PLR and PIPR were reported to change with circadian phase in strictly controlled conditions [13,14] but were not systematically associated with sleepiness ([31] but see [13]). A recent study reported that the association between steady-state pupil size under constant low light and sleepiness was only apparent during the circadian day while well-rested or sleep-deprived, but not over the circadian night [21]. Other studies failed to find any significant changes in steady-state pupil size under constant light level related to time-of-day, circadian phase, or sleep-wake history ([32], baseline pupil size prior to PLR investigation in [13,14]).

Discrepancies between studies likely arise from methodological differences, which may have biased the effects of interest, and/or from differences in statistical power. Sleepiness was for instance measured subjectively through the Likert scale or visual analogue scales, or objectively through electrophysiology and performance to sustained attention task (psychomotor vigilance task—PVT) while ambient light level were not always constant and/or low [15,20,22,31].

Here, we asked whether steady-state pupil size and its variability under constant dim ambient light changed during prolonged wakefulness in a strictly controlled protocol, including a sample of 20 young men. We hypothesized that steady-state pupil size variations and its variability would be related to subjective sleepiness and to typical objective behavioral and electrophysiological correlates of alertness. Lastly, since our protocol also included subjective measurements of affective state, we

explored whether these affective measures were related to steady-state pupil size and variability in pupil size.

## 2. Results

Twenty healthy young men (22.9 ± 2.7 years old, Table 1) completed a 29 h sleep deprivation protocol conducted under constant dim-light (<5 lux at eye level) in strictly controlled constant routine conditions (Figure 1). Steady-state pupil size and its variability (i.e., fast and low amplitude variations in steady-state pupil size) were assessed on 12 occasions throughout the protocol while fixating a dot for 2 min and suppressing blinks. Hourly saliva samples were collected for subsequent melatonin assays to determine dim-light melatonin onset (DLMO), which was set to circadian phase 0° and to which all collected data were realigned. Time is therefore expressed in degrees from DLMO, with 15° corresponding to 1 h.

**Table 1.** Sample demographics, questionnaires scores (mean ± SD).

| | |
|---|---|
| *N* | 20 |
| **Age (y)** | 22.9 (2.7) |
| **Ethnicity** | Caucasian |
| **BMI (kg/m$^2$)** | 22.1 (2.2) |
| **Anxiety level (BAI)** | 1.4 (2.2) |
| **Mood (BDI-II)** | 1.1 (1.9) |
| **Caffeine (cups/day)** | 0.4 (0.5) |
| **Alcohol (doses/week)** | 3.2 (3) |
| **Sleep quality (PSQI)** | 3.8 (0.7) |
| **Daytime propensity to fall asleep (ESS)** | 3.5 (2.9) |
| **Chronotype (HO)** | 52.9 (5.1) |
| **Sleep time (hh:min, sleep diary)** | 23:37 (44 min) |
| **Wake time (hh:min, sleep diary)** | 7:33 (45 min) |
| **Sleep time (hh:min, actigraphy)** | 23:31 (46 min) |
| **Wake time (hh:min, actigraphy)** | 7:31 (47 min) |
| **Average clock time of dim-light melatonin onset (hh:mm)** | 21:07 (77 min) |

Anxiety was measured by the 21-item Beck Anxiety Inventory (BAI ≤ 14) [33]; mood by the 21-items Beck Depression Inventory II (BDI-II ≤ 14) [34]; sleep quality by the Pittsburgh Sleep Quality Index Questionnaire (PSQI ≤ 7) [35]; daytime propensity to fall asleep in non-stimulating situations by the Epworth Sleepiness Scale (ESS ≤ 11) [36]; chronotype by the Horne-Östberg Questionnaire (<42: evening types; 42–58: intermediate types; >58: morning types) [37].

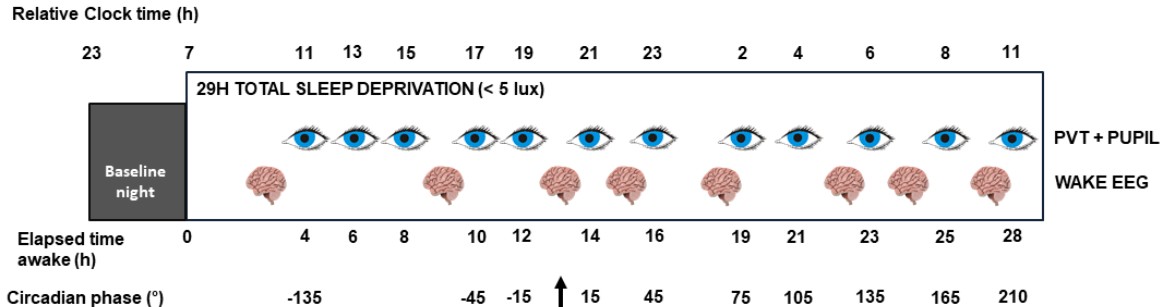

**Figure 1.** Experimental protocol. After an 8-h baseline night of sleep under polysomnographic recording, participants (*N* = 20) underwent 29 h of sustained wakefulness in constant routine conditions in dim-light (<5 lux at eye level). Pupil size was assessed 12 times immediately after a test battery including the psychomotor vigilance task (PVT). Spontaneous waking electroencephalographic (EEG) recordings were acquired 8 times. Saliva samples and subjective scores for sleepiness, fatigue and affective state were collected hourly. Time is expressed in expected circadian phase (degrees °; 15° = 1 h) and equivalent elapsed time awake (h). Relative clock time displayed here is for a participant with a 24:00–08:00 sleep-wake schedule. Arrow indicates expected time of dim-light melatonin onset (circadian phase 0°).

Statistical analyses revealed a significant change in steady-state pupil size (mm$^2$) across circadian phases (Figure 2A—GLMM; main effect of circadian phase—$F_{12,179.5}$ = 4.15; $p < 0.0001$; $R^2_{\beta*}$ = 0.22). Subsequent post hoc comparisons indicated that steady-state pupil size was larger around DLMO compared to data collected before and after (−15° > 165° & 195°, $p_{corrected} < 0.04$; 15° > −135°, −75°, 135°, 165°, 195° & 225°, $p_{corrected} < 0.025$; 45° > 165°, 195° & 225°, $p_{corrected} < 0.02$), while steady-state pupil size measured at the end of the protocol was significantly smaller than data collected earlier (165° & 195° < −135°, −15°, 15°, 45° & 75°, $p_{corrected} < 0.05$; 195° < 105°, $p_{corrected}$ = 0.006). Steady-state pupil size did not significantly vary before DLMO (<0°). Statistical analysis of pupil size variability led to no significant variation with circadian phase (Figure 2B—GLMM; main effect of circadian phase—$F_{12,162.5}$= 0.87; $p$ = 0.57).

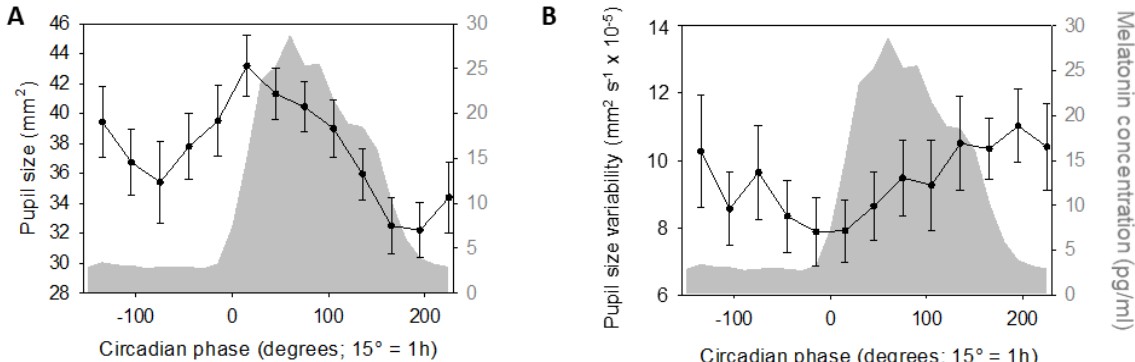

**Figure 2.** Time course of pupil measures during 29 h of sustained wakefulness (mean ± sem). (**A**) Steady-state pupil size (mm$^2$) significantly varied with circadian phase. (**B**) Pupil size variability (mm$^2$ s$^{-1}$) did not yield significant variation with circadian phase. Refer to main text for outputs of statistical tests. Average melatonin profile is displayed in grey. Time course is expressed relative to individual dim light melatonin onset (DLMO = phase 0°; 15° = 1 h).

Subjective sleepiness, collected through a Likert scale [38], was also measured hourly (and before psychomotor vigilance task) during the protocol and showed typical significant variations with circadian phase (Figure 3A—GLMM; main effect of circadian phase—$F_{12,178.3}$ = 9.45; $p < 0.0001$; $R^2_{\beta*}$ = 0.39), with low values during the first biological day and a significant increase mainly during

the biological night (−135° & −105° < (−75°)–225°, $p_{corrected}$ < 0.002; −75°, −45° & −15° < 75°–195°, $p_{corrected}$ < 0.03; 15° < 75°–165°, $p_{corrected}$ < 0.04; 45° < 135°, $p_{corrected}$ = 0.02). Likewise, subjective fatigue, assessed hourly through a visual analogue scale (VAS) (and immediately prior to pupil measurements), showed significant variations with circadian phase (Figure 3B—GLMM; main effect of circadian phase—$F_{12,180.5}$ = 14.89; $p$ < 0.0001; $R^2_{\beta*}$ = 0.50) with a similar pattern to subjective sleepiness (−135° & −105° > 15°–225°, $p_{corrected}$ < 0.02; −75°, −45° & −15° > 45°–225°, $p_{corrected}$ < 0.002; 15° > 75°–195°, $p_{corrected}$ < 0.01; 45° > 105°–165°, $p_{corrected}$ < 0.01; 75° > 135°–165°, $p$ < 0.01; 165° < 195°, $p_{corrected}$ = 0.02).

Performance to the psychomotor vigilance task (PVT), which probes sustained attention [39], was assessed during a test battery conducted 12 times during the protocol, immediately prior to VAS and pupil size measures. Mean reaction time to the PVT also showed significant variation with circadian phase (Figure 3C—GLMM; main effect of circadian phase—$F_{12,183.1}$ = 3.82; $p$ < 0.0001; $R^2_{\beta*}$ = 0.2), with biological night measures slower than data collected during the first biological day (−135° < 105°–195°, $p_{corrected}$ < 0.04; −105°, −75°, −15°, 15°, 45°, 75° < 165°, $p_{corrected}$ < 0.05; −45° < 135°–165°, $p_{corrected}$ < 0.04). Electroencephalogram (EEG) recordings of spontaneous waking brain activity at rest were also collected 8 times during the protocol to extract EEG theta power (4.5–7.5 Hz), which is a well-accepted measure of alertness (more theta power reflecting lower alertness) [40]. Similar to subjective sleepiness and PVT mean reaction times, relative theta power showed its typical variation with circadian phase (Figure 3D—GLMM; main effect of circadian phase—$F_{12,164.8}$ = 2.54; $p$ = 0.004; $R^2_{\beta*}$ = 0.16), with relative power being higher during the biological night when compared to the biological day (−135°, −105° & 15° < 135°–195°, $p_{corrected}$ < 0.04; −75° −45° & −15° < 105°–195°, $p_{corrected}$ < 0.05).

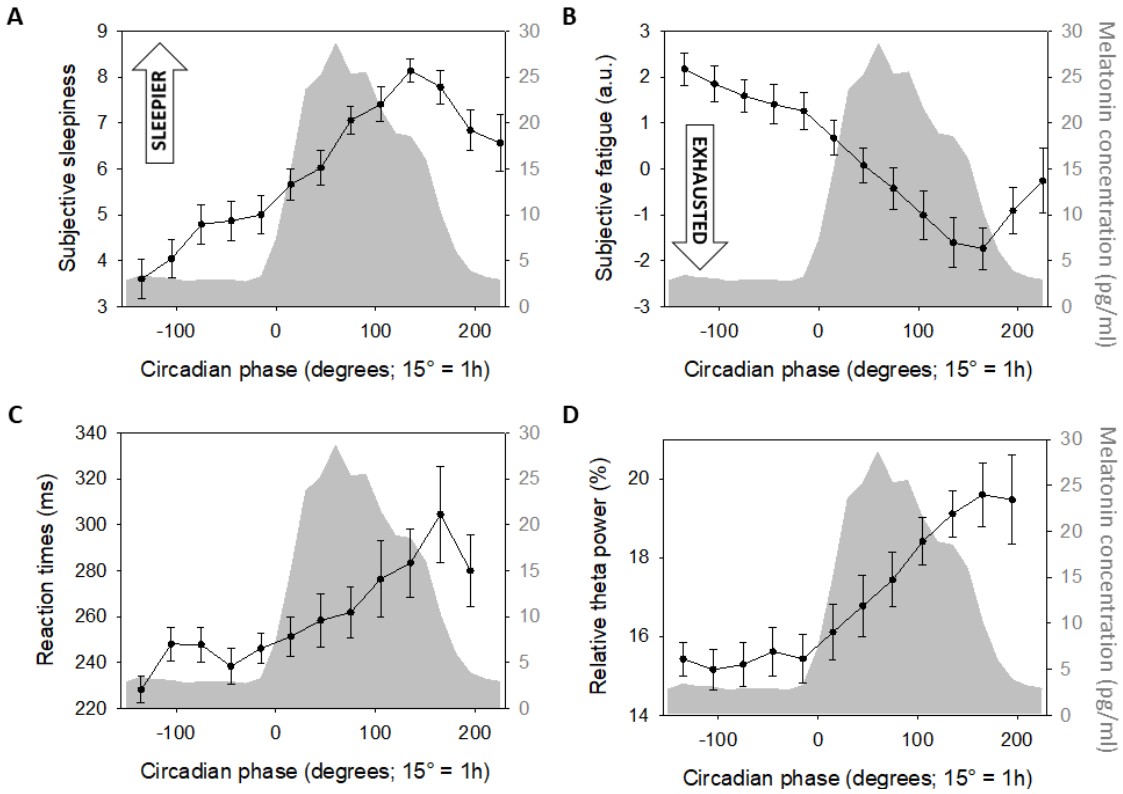

**Figure 3.** Time course of subjective sleepiness and fatigue, objective alertness and attention (mean ± sem). (**A**) Subjective sleepiness (Likert scale). (**B**) Subjective fatigue (visual analogue scale, arbitrary units—a.u.). (**C**) Mean reaction times during PVT. (**D**) Relative EEG theta power (4.5–7.5 Hz). All four variables significantly varied with circadian phase. Refer to main text for outputs of statistical tests. Average melatonin profile is displayed in grey. Time course is expressed relative to individual dim light melatonin onset (DLMO = phase 0°; 15° = 1 h).

We then sought for associations between pupil measures (independent variable together with circadian phase) and subjective sleepiness, subjective fatigue, PVT mean reaction times, and relative theta power taken separately as the dependent variable. Analyses including steady-state pupil size or its variability yielded no significant associations (Table 2), except for a significant linear association between steady-state pupil size and PVT mean reaction time, and between pupil size variability and subjective fatigue in interaction with circadian phase (Figure 4). This implies that the link between pupil size variability and subjective fatigue depends on the circadian phase. Qualitative inspection of the simple regressions computed considering each circadian phase separately yielded no clear picture with phases indicating that feeling more exhausted seems related to less variability (phases −135°, −105°, 15°, 45°, 75°, 105°, 165°, 225°), interleaved with phases when feeling more fresh seems associated with less variability (phases −75°, −45°, −15°, 135°, 195°) (Figure 4B,C).

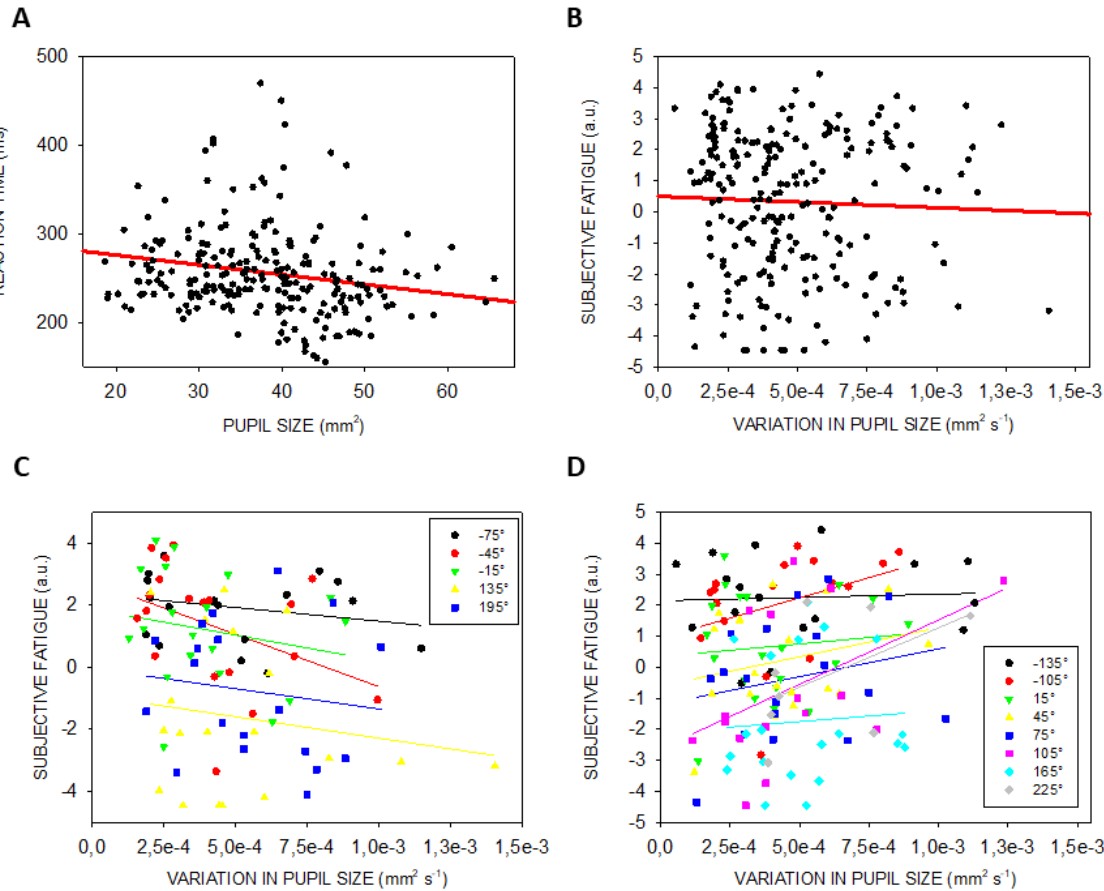

**Figure 4.** Linear associations between steady-state pupil size and PVT performance and between pupil size variability and subjective fatigue. (**A**) Regression between steady-state pupil size and mean reaction time during a PVT across all circadian phases. (**B**) Regression between pupil variability and subjective fatigue (assessed with a VAS) irrespective of circadian phase. (**C**) Regressions, considering each circadian phase separately, indicating that increase subjective fatigue is associated with decrease pupil size variability. (**D**) Regressions, considering each circadian phase separately, indicating that increase subjective fatigue is associated with increase pupil size variability. Phases included in panels **B** and **C** are indicated in the insets. Refer to main text for outputs of statistical tests. Regressions were used for visual display only, and not as a substitute of the full GLMM statistics presented in Table 2. GLMMs were set according to dependent variables distribution.

**Table 2.** Statistical outputs of the GLMMs seeking for associations between alertness and attention measures and pupil size measures.

| | | *Steady-State Pupil Size* | Circadian Phase | Steady-State Pupil Size x Circadian Phase |
|---|---|---|---|---|
| *Dependent variables* | **Subjective Sleepiness** | $F_{1,199.7} = 0.03$ <br> $p = 0.87$ | **$F_{12,158} = 2.65$** <br> **$p = 0.03$** <br> **$R^2_{\beta*} = 0.17$** | $F_{12,157} = 1.23$ <br> $p = 0.27$ |
| | **PVT mean reaction times** | **$F_{1,178.8} = 5.69$** <br> **$p = 0.02$** <br> **$R^2_{\beta*} = 0.03$** | $F_{12,165.1} = 0.98$ <br> $p = 0.47$ | $F_{12,165.1} = 0.65$ <br> $p = 0.8$ |
| | **Relative theta power** | $F_{1,147.4} = 1.19$ <br> $p = 0.28$ | $F_{12,143.7} = 1.31$ <br> $p = 0.22$ | $F_{12,143} = 1.00$ <br> $p = 0.46$ |
| | **Fatigue** | $F_{1,196.4} = 0.03$ <br> $p = 0.86$ | **$F_{12,165.2} = 2.93$** <br> **$p = 0.001$** <br> **$R^2_{\beta*} = 0.18$** | $F_{12,164.8} = 1.14$ <br> $p = 0.33$ |

**Table 2.** *Cont.*

| | | *Pupil size variability* | Circadian phase | Pupil size variability x circadian phase |
|---|---|---|---|---|
| *Dependent variables* | **Subjective sleepiness** | *$F_{1,187.6} = 2.75$* <br> *$p = 0.099$* | **$F_{12, 159.1} = 3.58$** <br> **$p < 0.001$** <br> **$R^2_{\beta*} = 0.21$** | $F_{12,157.1} = 0.99$ <br> $p = 0.46$ |
| | **PVT mean reaction times** | $F_{1,10,65} = 0.2$ <br> $p = 0.66$ | $F_{12,163.1} = 0.9$ <br> $p = 0.55$ | *$F_{12,150.3} = 1.64$* <br> *$p = 0.085$* |
| | **Relative theta power** | $F_{1,173.9} = 1.22$ <br> $p = 0.27$ | *$F_{12,144.4} = 1.80$* <br> *$p = 0.059$* | $F_{12,142} = 1.21$ <br> $p = 0.29$ |
| | **Fatigue** | *$F_{1,183.7} = 1.61$* <br> *$p = 0.21$* | **$F_{12,167.6} = 8.49$** <br> **$p < 0.0001$** <br> **$R^2_{\beta*} = 0.38$** | **$F_{12,166.1} = 2.54$** <br> **$p = 0.004$** <br> **$R^2_{\beta*} = 0.16$** |

Each dependent variable was included in a separate GLMM including pupil measure, circadian phase, and their interaction. Degrees of freedom were computed following Kenward-Roger method. Significant linear links are in bold and statistical trends ($p < 0.1$) are in italic. $R^2_{\beta*}$ represents semi-partial $R^2$ as proposed by [41] (see methods). GLMMs were set according to dependent variables distribution.

In a last set of exploratory analyses, we considered subjective affective measures. These were acquired hourly together with subjective fatigue, right after the PVT and right before pupil measurement, and included 5 items: motivation, joy, stress, anguish, and sociability. We first confirmed that all affective dimensions significantly varied with sleep-wake history [42] (Figure 5; main effect of circadian phase; *Motivation*: $F_{12,179.4} = 5.54$, $p < 0.001$, $R^2_{\beta*} = 0.27$; *Joy*: $F_{12,181.8} = 5.46$, $p < 0.001$, $R^2_{\beta*} = 0.26$; *Stress*: $F_{12,178} = 3.66$, $p < 0.001$, $R^2_{\beta*} = 0.2$; *Anguish*: $F_{12,178} = 2.29$, $p = 0.01$, $R^2_{\beta*} = 0.13$; *Sociability*: $F_{12,173.2} = 2.44$, $p = 0.006$, $R^2_{\beta*} = 0.14$). Post hoc yielded similar results across all measures with better affect during the biological day as compared to the biological night [(main post hocs: *Motivation/Joy/Stress* ($-135° > 45°–225°$, $p_{corrected} < 0.05$; $-105° > 135°–225°$, $p_{corrected} < 0.05$; $-75°$, $-45°$, $-15°$, $> 135°–165°$, $p_{corrected} < 0.05$; $165° < 195°$, $p_{corrected} < 0.02$); *Anguish/Sociability* ($-135°$, $-105° > 135°–165°$, $p_{corrected} < 0.05$)].

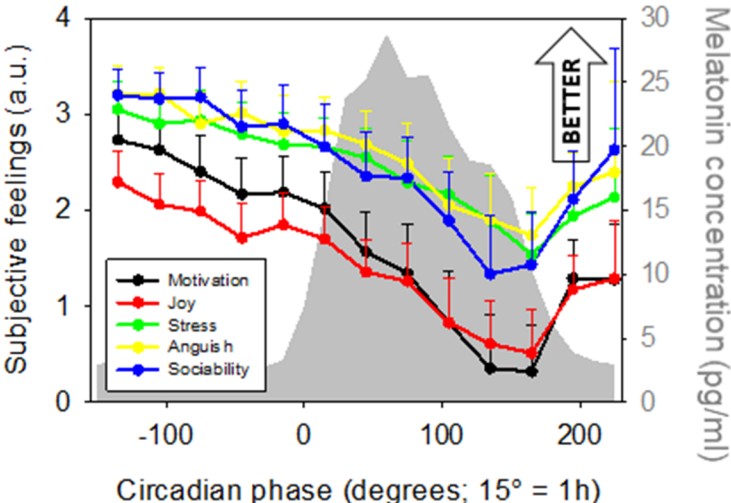

**Figure 5.** Time course of subjective affective dimensions (mean ± sem; arbitrary units—a.u.). All five variables significantly varied with circadian phase. Refer to main text for outputs of statistical tests. Average melatonin profile is displayed in grey. Time course is expressed relative to individual melatonin onset (DLMO = phase 0°; 15° = 1 h).

We then asked whether steady-state pupil size or variability in pupil size could be associated with either one of the affective dimensions probed during the protocol. Table 3 indicates that motivation is significantly related to steady-state pupil size during prolonged wakefulness, with higher motivation linked to larger pupils (Figure 6A). In addition, lower sociability and higher anguish were significantly associated with higher pupil size variability (Figure 6B,C).

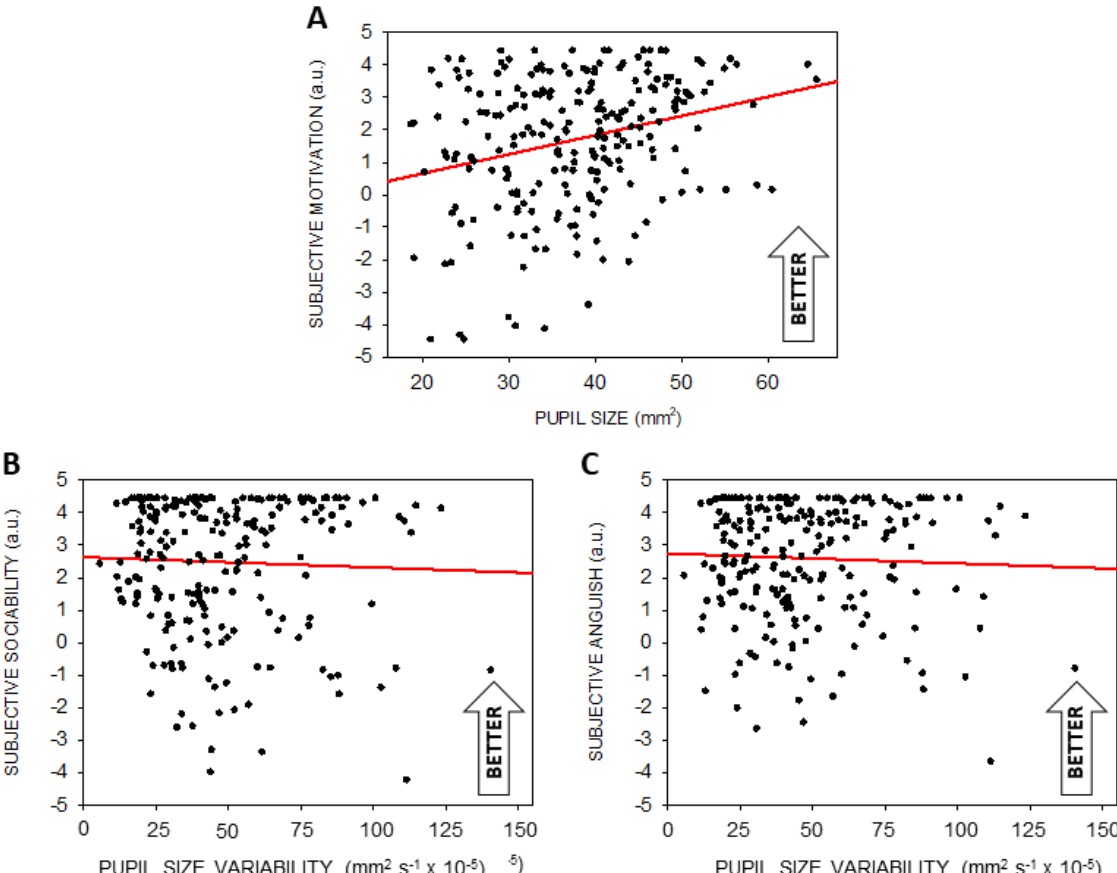

**Figure 6.** Significant linear associations between pupil measures and subjective affective dimensions (arbitrary units—a.u.). (**A**) Significant linear associations between steady-state pupil size and motivation. (**B**) Significant linear associations between pupil size variability and sociability. (**C**) Significant linear associations between pupil size variability and anguish. Refer to main text for outputs of statistical tests. Regressions were used for visual display only, and not as a substitute of the full GLMM statistics presented in Table 3. GLMMs were set according to dependent variables distribution.

**Table 3.** Statistical outputs of the GLMMs seeking for association between subjective affective dimensions and pupil size measures.

| | | Steady-State Pupil Size | Circadian Phase | Steady-State Pupil Size x Circadian Phase |
|---|---|---|---|---|
| | Motivation | $F_{1,196.7} = 4.46$ $p = 0.04$ $R^2_{\beta*} = 0.022$ | $F_{12,162.2} = 2.79$ $p = 0.002$ $R^2_{\beta*} = 0.17$ | $F_{12,161.7} = 1.62$ $p = 0.09$ |
| | Joy | $F_{1,194.8} = 2.81$ $p = 0.095$ | $F_{12,164.5} = 1.90$ $p = 0.04$ $R^2_{\beta*} = 0.12$ | $F_{12,164} = 1.24$ $p = 0.26$ |
| *Dependent variables* | Stress | $F_{1,172.9} = 1.20$ $p = 0.27$ | $F_{12,160.5} = 1.60$ $p = 0.096$ | $F_{12,160.4} = 1.00$ $p = 0.45$ |
| | Anguish | $F_{1,199.5} = 0.87$ $p = 0.35$ | $F_{12,162.4} = 1.61$ $p = 0.094$ | $F_{12,161.5} = 1.48$ $p = 0.14$ |
| | Sociability | $F_{1,198.4} = 0.68$ $p = 0.41$ | $F_{12,161.2} = 2.19$ $p = 0.01$ $R^2_{\beta*} = 0.14$ | $F_{12,160.2} = 1.52$ $p = 0.12$ |

**Table 3.** *Cont.*

| | | *Pupil size variability* | Circadian phase | Pupil size variability x circadian phase |
|---|---|---|---|---|
| *Dependent variables* | Motivation | $F_{1,185.5} = 0.91$ $p = 0.34$ | $\mathbf{F_{12,165.2} = 1.99}$ $\boldsymbol{p = 0.03}$ $\boldsymbol{R^2_{\beta*} = 0.13}$ | $F_{12,163.2} = 0.77$ $p = 0.68$ |
| | Joy | $F_{1,182.6} = 2.36$ $p = 0.13$ | $\mathbf{F_{12,165.2} = 3.34}$ $\boldsymbol{p = 0.0002}$ $\boldsymbol{R^2_{\beta*} = 0.2}$ | $F_{12,163.1} = 1.23$ $p = 0.27$ |
| | Stress | $F_{1,161.1} = 0.71$ $p = 0.40$ | $F_{12,161.9} = 0.76$ $p = 0.69$ | $F_{12,161.3} = 0.30$ $p = 0.99$ |
| | Anguish | $\mathbf{F_{1,193.9} = 4.81}$ $\boldsymbol{p = 0.03}$ $\boldsymbol{R^2_{\beta*} = 0.024}$ | $\mathbf{F_{12,165.2} = 1.90}$ $\boldsymbol{p = 0.04}$ $\boldsymbol{R^2_{\beta*} = 0.12}$ | *$F_{12,162.4} = 1.68$* *$p = 0.076$* |
| | Sociability | $\mathbf{F_{1,189.6} = 4.02}$ $\boldsymbol{p = 0.046}$ $\boldsymbol{R^2_{\beta*} = 0.021}$ | $F_{12,165.2} = 1.29$ $p = 0.23$ | $F_{12,162.6} = 0.84$ $p = 0.61$ |

Each dependent variable was included in a separate GLMM including pupil measure, circadian phase and their interaction. Degrees of freedom were computed following Kenward-Roger method. Significant linear links are in bold and statistical trends ($p < 0.1$) are in italic. $R^2_{\beta*}$ represents semi-partial $R^2$ as proposed by [41] (see methods). GLMMs were set according to dependent variables distribution.

## 3. Discussion

In line with our hypothesis, we report a robust change in steady-state pupil size during 29 h of continuous wakefulness in a relatively large sample ($N = 20$) of healthy young men maintained in dim-light in strictly controlled constant routine conditions. Contrary to our expectations, we did not detect significant changes in pupil size variability, i.e., fast and low amplitude variations in steady-state pupil size. Also contrary to our hypotheses and to previous findings, neither steady-state pupil size nor its variability were significantly related to subjective sleepiness and electrophysiological correlates of alertness, but, as anticipated, pupil size variation was associated with PVT performance. Our data support therefore the notion that, compared with its variability, steady-state pupil size is strongly influenced by the concomitant changes in sleep need and circadian phase, which cannot be dissociated in a constant routine protocol. In the context of our experiment, pupil measurements are not related to subjective or objective measures of sleepiness/alertness, but to performance of a sustained attention task, closely related to sleepiness/alertness.

The reported variations in steady-state pupil size with time awake nicely reproduce a recent study, also carried out in constant dim-light and constant routine conditions [21]. Steady-state pupil size variations could be driven by the central nervous system, local retinal processes, or both. In mammal, as well as other animal Classes, the master circadian clock consists of the suprachiasmatic nuclei (SCN) located within the hypothalamus [43,44]. Activity synchrony between SCN neurons sets a signal that orchestrates peripheral clocks in the brain, including the retina [45], and other organs [46] as well as the clock of each and every cells of the organism [43]. The SCN indirectly connect the pineal, via spinal ganglions, to drive melatonin secretion, and the LC, through the dorsolateral hypothalamic (DLH), to modulate its activity [47,48]. The SCN may therefore influence steady-state pupil size through these paths. Sleep-wake history also affects global brain state notably through changes in the neurochemical environment of brain cells associated with sleep homeostasis: increase in extracellular concentration of compounds such as adenosine, different ions and glutamate have been directly related to increased sleep need [49–51]. The latter chemical changes could also be present locally in the retina and drive local changes in pupil size. To our knowledge, evidence of local retinal mechanisms related to sleep-wake history driving retinal physiology remains nonexistent, however.

In contrast, the existence of a local circadian clock is well established and, contrary to most other peripheral clocks, is self-sustained, i.e., it persists for long periods even when disconnected from the SCN [45,52]. The site of this retinal clock is not definitely established and seems to include classical

photoreceptors themselves (i.e., rods and cones) as well as retinal ganglions cells [45,52]. The retina shows for instance strong variations in local melatonin concentration [45,52]. Retinal circadian rhythms allow the organism to anticipate and adapt to the >1 million-fold changes in light intensity during a 24-h period, optimizing visual function for each photic situation [45,52]. The significant changes in steady-state pupil size we observe may therefore be in part anticipatory. We observe an increase in pupil size during the day, up to the so-called wake maintenance zone [53], potentially to adapt to the reduction in environmental light. Yet, steady-state pupil variation does not appear as a pure sine wave, but rather a combination of a putative circadian and sleep homeostasis influence. Overall, it is likely that both local and global changes arising from circadian processes and sleep homeostasis drive variation in pupil size, and we are in no position to isolate their respective contribution.

According to our results, steady-state pupil size is therefore among the retinal parameters modified with time-of-day, sleep-need, and circadian phase, together with levels of visual pigments, GABA, dopamine, melatonin, visual sensitivity (all reviewed in [45,52]), PLR [13,14], blink rate, or eye opening [54]. The variation in visual sensitivity and PLR may be of importance in the context of our study. Even though we carefully maintained a constant dim-light level (<5 lux) throughout the protocol, a change in visual sensitivity could drive visual adaptation and potentially changes in pupil size. Likewise, variations in the sensitivity of the non-image-forming photoreception system could affect PLR and steady-state pupil size [55,56]. Changes in sensitivity could be originating from ipRGCs expressing melanopsin, as they also contribute to adaptation to the ambient light levels [11].

We find that steady-state pupil size is significantly associated with PVT performance as indexed by mean reaction times. This suggests that variations in sustained-attention, which is closely related to vigilance and alertness [57], during prolonged wakefulness is closely related to pupil size variation, reminiscent of many prior studies [20,24,28,29,31,32]. Yet, steady-state pupil size was not significantly associated with measures typically used in studies focused on sleep-wake history, including EEG theta power and subjective sleepiness. The reason for this apparent discrepancy is unclear, particularly given that the time-courses of all three measures appear qualitatively similar at the group level. Our results should be considered within the context of our study. First, pupil recordings were 2 min long. This is short compared to other studies that reported increased pupil size variability within a recording session lasting about 10 min or more [20,23,58,59]. Fixating a dot for 2 or 10 min is very different in term of vigilance/sustained attention challenge [60] so that pupil size variability may be more strongly associated with alertness/sleepiness during longer recordings. Second, all measures were not collected simultaneously: subjective sleepiness and PVT performance were assessed, respectively ~15 min and ~5 min prior to pupil measurement, while EEG recordings were even further apart (cf. Figure 1). This may be important, if the associations between pupil measures and measures such as sleepiness are very short-lived and very sensitive to phasic variations, arising for instance from transient changes in motivation or effort [20]. In addition, our study only includes healthy young men aged 18 to 30 years when age and sex have been suggested to influence light sensitivity [61,62]. One should bear in mind that the associations between pupil measures and subjective sleepiness and objective alertness may just be weak, at least compared to other parameters such as PVT performance and not easy to put forward even with a sample of 20 individuals. Semi-partial R-square of the association between pupil measures and PVT performance were, however, relatively low in our data set ($R^2_{\beta*} \sim 0.03$).

We further find that, despite its lack of significant changes across the protocol, pupil size variability was associated with fatigue, as previously suggested [3]. Since LC phasic activity may drive pupil size variability [9], this finding may reflect a link between fatigue and LC activity, as previously proposed [63]. The absence of link between pupil measures and subjective sleepiness may appear as a surprise given the present link with fatigue, and given previous reports [20,24,28,29,31,32]. Yet, fatigue is distinct from sleepiness because it encompasses a cognitive dimension (one may be cognitively fatigued while not being sleepy because of lack of sleep, and vice-versa) [3]. The distinction between fatigue and sleepiness may not be clear across studies or even across study participants and may explain some inconsistencies in the links between the pupil and sleep-wake history [64]. Sleepiness

and fatigue also differed in the mean to assess them during our protocol, respectively through a Likert scale and a VAS. How this may affect measurements is unclear. One should still take careful considerations of how subjective sleep-wake state is collected. Finally, in our analyses, the link between pupil size variability and fatigue changed with circadian phase, but without clear distinction between periods with a positive or negative correlation. This is similar to a recent study showing that the link between pupil size and subjective sleepiness was only evident during the circadian day and over the circadian night [21]. In the present study however, we did not group circadian phases according to day and night and no portion of the circadian cycle seems to clearly stand out in its link with pupil size variability. This will require further investigations which should also assess whether variation in pupil size variability is related to changes in accommodation behavior (while fixating the black dot at the wall in the present study), which may be more difficult over the different parts of the protocol.

Our exploratory analyses further suggest that pupil size variability is related to sociability and anguish across the entire protocol. Again, considering the role of the LC in emotional processes (including anxiety, mood) [65], this finding could be related to LC phasic activity, but other regions such as the amygdala or the hypothalamus, which is key to emotions and impinges on sympathovagal balance, are as likely. Even if data distribution is accounted in the GLMM we computed, the scores collected were always very positive (toward "friendly" and "relaxed") with what resembles a ceiling effect. Finally, steady-state pupil size seems to be closely related with motivation, in line with other reports [2,3]. Motivation is indeed an important aspect of one's condition that may override, at least in part, sleep-wake signal arising from circadian and/or sleep homeostasis processes [66]. Given the relatively low semi-partial effect size reported in Table 3, the results linking the pupil to affective dimension should be considered with caution.

## 4. Methods

### 4.1. Participants

The study was approved by the Ethics Committee of the Medicine Faculty of the University of Liège. Participants gave their written informed consent. 24 healthy Caucasian men (18–30 years old) were enrolled. Women were excluded because of potential sex influence on other measures of interest of the protocol that are not reported here [67–69]. Exclusion criteria included: (1) Body Mass Index (BMI) ≤18 and ≥25; (2) psychiatric history, severe trauma, sleep disorders; (3) addiction, chronic medication; (4) smokers, excessive alcohol (>14 doses/week) or caffeine (>3 cups/day) consumption; (5) night shift workers during the last year; (6) transmeridian travel during the last two months; (7) anxiety or depression; (8) poor sleep quality; (9) excessive self-reported daytime sleepiness.

One participant was excluded due to melatonin phase-delay >6 h compared to the remainder of the sample. Three participants did not have enough data passing quality control (see below) and were excluded. Thus, data presented here include 20 participants. Table 1 summarizes the demographic characteristics of the final study sample.

### 4.2. Experimental Protocol

Participants completed a screening night of sleep to exclude major sleep disorders. During the 7 days preceding the study, they kept a regular 8 h sleep-wake schedule (+/−15 min; verified using wrist actigraphy—Actiwatch, Cambridge Neurotechnology, UK, and sleep diaries). Participants were requested to abstain from all caffeine and alcohol-containing beverages for 3 days preceding the study.

For the experiment *per se*, participants were maintained in dim-light for 6 h (<5 lux), prior to sleeping for 8 h at their habitual bedtime in complete darkness. They then remained awake for 29 h of sustained wakefulness period under constant routine conditions (i.e., light <5 lux, temperature ~19 °C, regular isocaloric liquid meals and water, semi-recumbent position, no time-of-day information, sound-proofed rooms). These conditions aim at minimizing external and internal factors masking circadian rhythmicity [70]. Spontaneous quiet eyes-open waking EEG recordings were acquired

8 times during the prolonged wakefulness period (11:00, 17:00, 21:00, 23:00, 02:00, 06:00, 08:00, 11:00, for a subject sleeping from 24:00 to 08:00; Figure 1). Psychomotor vigilance task assessment (PVT) followed by pupil measurement were carried out 12 times during the protocol in between waking EEG recordings (12:00, 14:00, 16:00, 18:00, 20:00, 22:00, 24:00, 03:00, 05:00, 07:00, 09:00, 12:00). Subjective perception (sleepiness, affective dimension) was assessed hourly using as well saliva samples for subsequent melatonin assays.

### 4.3. Pupil Measures

Pupil size was recorded using an infra-red light-based eye-tracking device (Arrington Research; sampling rate: 90 Hz). Participants were seated in bed (upright position) and rested their head on a chin rest while fixating a black dot on the wall for 2 min and suppressing blinks. Each recording was preceded by a 10 s recording of an 8-mm diameter black sticker placed on the closed eyelid for subsequent normalization of pupil measurement according to a reference of known diameter. Blinks and bad quality data were automatically removed: data were excluded if the ratio between pupil width and height was <0.8 or >1.2 or if arbitrary unit pupil diameter was extremely small or high. Visual inspection of the automatic data rejection was subsequently carried out. Recordings with >30% of excluded data were discarded. Participants with >50% of missing recordings or with >3 consecutive missing recordings were excluded from the analyses (3 participants were discarded cf. above). Within the 20 participants that passed these quality checks, 2.5% ± 3.3% (mean ± SD) of data was missing. Time series were then smoothed using Robust LOWESS linear fit in Matlab 2015b (smooth function, 'lowess' method, 270 data point span, i.e., 3 s span).

Pupil size was determined based on pupil width and height according to the formula for elliptic surface [$\pi$*(width/2)*(height/2)]. Two measures of interest were extracted from the resulting time series: steady-state pupil size, computed as the average size over the 2 min recording, and pupil size variability, computed as the square root of the square sum of second-to-second differences. Steady-state pupil size was normalized according to the surface of the black sticker recording prior to each pupil measurement (10 s average of elliptic surface of the sticker). Sessions with >10 s consecutive seconds of missing data were not included when considering pupil size variability.

### 4.4. Subjective Sleepiness or Fatigue

We administered hourly Karolinska Sleepiness Scale (KSS) [38], consisting in a 9 position Likert scale (1 = very alert; 9 = very sleepy difficulties remaining awake) answering to the question "how are you feeling now?". KSS was administered at the start of each cognitive batteries, i.e., ca. 15 min prior to pupil measurements. Subjective fatigue score was collected hourly using a visual analog scale (VAS) (from left: fresh, to right: exhausted) which was included in a series of 6 VAS (the other 5 VAS focusing on affective dimension). These VAS immediately followed the PVT, i.e., <5 min prior to pupil measurements and required participants to answer to the same question as for the KSS ("how are you feeling now?"). When no test battery was planned in a given hour, all subjective measures were collected one after the other (KSS then 6 VAS) following one-hour interval with preceding and following measurements.

### 4.5. Subjective Affective Dimensions

In addition to fatigue, hourly VAS included 5 affective item: motivation (from left: motivated, to right: demotivated), joy (from left: euphoric, to right: depressed), stress (from left: relaxed, to right: stressed), anguish (from left: calm, to right: anguished), sociability (from left: friendly, to right: irritable).

### 4.6. Psychomotor Vigilance Task (PVT)

Following subjective sleepiness assessment, participants performed three tasks which will not be further presented here (sustained attention reaction time task—serial 2 digit number addition;

2 min [71]; visual 2-back task—3 min; visual 3-back task—3 min [72]), and then the PVT. They were required to press a computer space bar as soon as an auditory signal occurred (presented at a random interval of 3–7 s [73]). We opted for an auditory version, because it would lead to fewer lapses of vigilance and potential micro-sleep episodes, which would have biased our results [74]. The PVT lasted 5 min [75]. Performance was inferred from the mean reaction times following removal of trials with anticipations (<100 ms) or errors (>3000 ms) [76].

### 4.7. Spontaneous Waking EEG

Spontaneous quiet eyes-open waking EEG (WEEG) was recorded using a 60-channel EEG (+2 EOG) amplifier (Eximia; Helsinki, Nexstim). Participants were instructed to fixate on a black dot for two minutes while relaxing and suppressing blinking. Data preprocessing was performed using SPM12 (https://www.fil.ion.ucl.ac.uk/spm/). Continuous EEG recordings were band-pass filtered between 0.1–500 Hz and resampled from 1450 to 500 Hz. Data were then manually and visually scored offline for artefacts and micro-sleep episodes (eye blinks, body movements, and slow eye movements), using FASST toolbox (http://www.montefiore.ulg.ac.be/~{}phillips/FASST.html). Power spectral densities were computed using a fast Fourier transform on artifact-free 4 s time windows, overlapping by 2 s, using the Welch's method (pwelch function in MATLAB 2011a) [77]. EEG activity was computed over frontal regions for theta (4.5–7.5 Hz) frequency bands over the entire 2 min recording.

### 4.8. Melatonin

Saliva samples were first placed at 4 °C, prior to centrifugation and congelation at −20 °C within 12 h. Salivary melatonin was measured by radioimmunoassay (Stockgrand Ltd., Guildford, UK), as previously described [78]. The limit of detection of the assay for melatonin was 0.8 ± 0.2 pg/mL using 500 μL volumes. Most samples were analyzed in duplicate. Estimation of individual's dim light melatonin onset (DLMO = phase 0°) was determined based on raw values. The 4 first samples were disregarded and maximum secretion level was set as the median of the 3 highest concentrations. Baseline level was set to be the median of the values collected from "wake-up time + 5 h" to "wake-up time + 10 h". DLMO was computed as the time at which melatonin level reached 20% of the baseline to maximum level (linear interpolation) [79].

### 4.9. Statistics

The circadian phase of all data points was estimated relative to individual DLMO (i.e., phase 0°, 15° = 1 h). All data points were resampled following linear interpolation every 30° (2 h) (Figure 1): −135°, −105°, −75°, −45°, −15°, 15°, 45°, 75°, 105°, 135°, 165°, 195° and 225°. Data were not extrapolated beyond 15° (i.e., 1 h), such that resampling at 225° of PVT data and of spontaneous activity EEG data could not be carried out and only includes data up to 195°. Effect of phase on spontaneous EEG activity was first assessed using the theoretical phases of the recording in the protocol (−150°, −60°, 0°, 30°, 75°, 135°, 165° and 210°) but correlational analyses were carried out with behavioral test phases. Data points situated at ±3 standard deviations of sample average at each circadian phase were defined as outliers and removed (up to two data points were removed per analyses).

Statistical analyses were performed with SAS version 9.4 (SAS Institute, Cary, NC, USA). Generalized linear mixed models (PROC GLIMMIX) were applied to compute all statistics following determination of the dependent variable distribution (using 'allfitdist' Matlab function). Subject (intercept) effect was included as random factor. Circadian phase was included as the repeated measure together with an autoregressive estimation of autocorrelation of order 1 [AR(1)], and the covariance structure specified the subject. In all GLMMs, degrees of freedom were estimated using Kenward-Roger's correction. If an interaction term was significant, simple effects were assessed using post-hoc contrasts (difference of least square means) adjusted for multiple testing with Tukey's procedure. Simple regressions were used for visual display only, and not as a substitute of the full GLMM statistics.

When analyzing the time course of a given variable (i.e., mean pupil size, variation in pupil size, PVT mean reaction times, subjective sleepiness, relative theta power, VAS outputs), GLMM model included circadian phase. When seeking for associations between pupil measures and behavior (PVT mean reaction times, subjective sleepiness, theta power, VAS outputs), GLMM models included circadian phase and the interaction between circadian phase and pupil measures.

We report semi-partial $R^2$ for each significant effect of interest as described in [41]. Generalization of the $R^2$ statistic to GLMMs is a difficult task and remains an unresolved problem, with several method proposed [41,80]. We opted for the approach proposed and validated in [41] because it allows for a simple computation of semi-partial $R^2$ as [Sum of Squares/(1 + Sum of Square)], with [Sum of Squares = NumDF * FValue/DenDF] (NumDF: numerator degrees of freedom; DenDF; denominator DF), provided that degrees of freedom are estimated using Kenward-Roger's methods.

## 5. Conclusions

We report a robust variation in pupil size during prolonged wakefulness that most probably arises for the combined influence of local and global circadian and sleep homeostasis processes. These findings support that sleep-wake history control, or at least monitoring, should be implemented in all protocols considering tonic or steady-state pupil size, and baseline correction should be included in studies or clinical examination focused on phasic pupil responses (e.g., to light exposure). The findings have indeed implications for clinical practice where pupil measures, including PLR and PIPR, may aid diagnosis or follow-up in eye or brain diseases, like glaucoma, multiple sclerosis, Seasonal Affective Disorders or Parkinson's disease [81]. Finally, our study suggest that sustained-attention performance, fatigue, and affective states may be monitored via pupil measures, but does not support that they constitute reliable markers of subjective sleepiness and objective EEG measures of alertness, at least in healthy young men, and in the conditions inherent to the present experiment.

**Author Contributions:** G.G., J.Q.M.L. and G.V. acquired the data. M.V.E., C.C.-O., and G.V. analysed pupil data. G.G., J.Q.M.L., G.V. analysed the other data. G.V. designed the experiment and wrote the paper. All authors edited the manuscript.

**Funding:** The study was funded by Wallonia Brussels International (WBI to G.G.), Fonds Léon Fredericq (FLF to G.G.), Fonds National de la Recherche Scientifique (FRS-FNRS, FRSM 3.4516.11, Belgium), Actions de Recherche Concertées (ARC 09/14-03) of the Fédération Wallonie-Bruxelles, University of Liège (ULiège), Fondation Simone et Pierre Clerdent, AXA Foundation, Fondation Médicale Reine Elisabeth (FMRE, Belgium), European Regional Development Fund (ERDF; Radiomed project), Wallonie-Bruxelles International (WBI to G.G.), and Walloon excellence in life sciences and biotechnology (WELBIO-CR-2010-06E, Belgium). M.V.E., G.V. are supported by the FNRS-Belgium.

**Acknowledgments:** We thank S. Chellappa, A. Claes, C. Degueldre, J. Devillers, B. Herbillon, P. Hanwotte, E. Lambot, B. Lauricella, P. Maquet for their help in different steps of the study.

**Conflicts of Interest:** The authors declare no competing financial interests.

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
