# Peer review of "Steady-State Pupil Size Varies with Circadian Phase and Sleep Homeostasis in Healthy Young Men"

_2624-5175, doi:10.3390/clockssleep1020021_

Reviewer 1 Report

General comments to authors:

The paper addresses the question whether steady-state pupil size changes in constant dim light conditions across 29 hours. The study design uses a well-controlled laboratory paradigm in 20 healthy young men, and the methods are sound. The authors align steady state pupil sizes to longer durations of prior sleepiness as well as to circadian oscillation of melatonin onset secretion. They conclude that the steady-state pupil size is impacted by both, circadian and sleep-dependent effects and is therefore not suitable as marker for sleepiness alone.

This is not a new finding at all, and has been shown previously by different groups. Even though the paper is well written and the data presentation is clear, there are some inconsistencies and lack of preciseness especially in the introduction and discussion section. Some references were not accurately cited or important references were not cited at all. The authors should make very clear that they did not measure functionality/reactivity of the pupil in response to a light stimulus, which would have been a better indicator for changes to measure extended wakefulness, and/or circadian variation, even though this has also been clearly shown previously.

Title: please revise – there is also a sleep-dependent modulation, the title should also contain …’steady-state pupil size in constant dim light.’…

Detailed comments to the authors:

Abstract:

Line 17 : should state… no statistically significant changes. The abstract should make clearer what exactly was found.

From the main manuscript text, there is no statistical significant association with steady state pupil size and subjective sleepiness. There is a significant interaction of pupil size variability and fatigue across circadian phases (please see my comment in the results section – a post-hoc test needs to reveal at which circadian phase the interaction occurred). There is a mix between pupil size and pupil size variability please revise and structure better the abstract.

Introduction:

·         Lines 26 ff: Please cite original and landmark work done by Lowenstein O. and Lowenfeld I. and not just a recent review. L & L, they were the first who systematically investigated pupil responses in animal and humans.

·         Line 35: Pupil size is not exclusively regulated by vasovagal tone – what about direct afferents from melanopsin-dependent ipRGCs via Edinger-Westphal nuclei and other afferents? Please revise the text accordingly.

·         Lines 46 ff: The research group around B. Wilhelm did also quite some work on the pupil size as a potential marker for sleepiness in humans – please cite at least one of their studies here too (e.g. Wilhelm et al. 1998 or 2009).  

·         Line 50 – please be explicit – do you mean the pupil response to light (i.e. pupil size during or after a light stimulus) or are you talking about baseline pupil’s size (i.e. steady state pupil size) under constant conditions. The former is a functional test of the pupil reactivity and its responses were used to defer photoreceptor contribution. The latter is more a continuous measure of pupil size – may be similarly to brain activity in the magnet resonance imaging when looking at resting-state activity compared to functional MRI, in response to a stimulus or to imposed activity. This is a weakness of the paper please revise throughout the manuscript that you talk about steady-state pupil size. The wording should be defined at the very beginning of the method section and referenced studies should be described more precisely, i.e. what was exactly measured and what was the outcome?

·         Line 52 ff: References 7 and 21 (Zele et al. Munch et al.) are not correctly referenced. In both studies the aim was not to show differences in steady-state pupil size but differences in pupil size in response to red and blue light stimuli (as a stimulation of rather cones or ipRGCs) across the 24-h cycle, there they did not “fail to show this”. Both studies found a circadian component, and both studies showed that transient pupil sizes to light stimuli became smaller in the course of the protocol. Please revise the sentence accordingly. Please see also the general comments to authors.

·         Line 56 ff: please be more precise – which behavioural and alertness correlates? There is no mention of circadian measures here – but in the results the authors first show circadian measures. Was this a post-hoc hypothesis?

Results:

Figure 3B: Y-axis what does a.u. mean?

Figure 3: is nice but has been reported by many others previously. The authors could consider omitting this Figure or showing it in the supplementary material. One possibility is to combine fatigue and pupil size variability in a new Figure 3 since there was a significant interaction between both.

Line 135: what was the post-doc for this? A significant interaction would allow you to test the effects of pupil variability for each time point…the following sentences lines 136 ff should be omitted or supported by stats.

Discussion:

Overall: The discussion needs to be shortened and better structured and focused on the discussion of the results. The final paragraph should be much more succinct and modest since there were no clear new findings.  

·         Please start the discussion with a small summary paragraph of the findings.

·         Please check the reference citations!

·         Please focus your discussion on the hypotheses, which were stated at the end of the introduction.

·         It is not clear why the authors stress the connection to LC, please revise and explain.

·         Lines 199: the authors report a change across circadian phase and state at the end of the praragraph that …”steady-state pupil size is strongly influenced by the concomitant changes in sleep need and circadian phase. “… This sentence needs to be revised in order to make clear the current protocol does not allow separating the two processes from each other.

·         Line 201 ff: This paragraph is too long and should be shortened and it should be focused more on the discussion of the results.

·         One problem with the changes in variability might also be due to changes in accommodation behavior while fixating the black dot at the wall. This is a task per se which probably changes with extended duration of wakefulness or due to different strategies between individuals. You may add this to the discussion, since there was a significant interaction between fatigue and pupil size variability. The latter seems greatest around circadian times of greatest exhaustion (see my comment to line 135 ff – what were the post-hocs and in which direction did they go?).

Methods:

Lines 320 ff: What was the percentage of excluded pupil recordings? Please report this in the revised version of the manuscript.

Lines 378 ff: Please report mean DLMO (+/- SD) in clock time.

Lines 413 ff: was the data normally distributed prior to generating R2 statistics? If not – which transformation was used?

References:

Please check the format of the references – there are many small inconsistencies.

Reviewer 2 Report

- expand on the affective state measurements in your study and what would they tell in relation to pupil size measurements; define the cognitive battery, how does EEG theta power inform about alertness (this can be mentioned in results section); 

- Figure 1: edit the figure so that the time scale would be equally distributed, you might even consider re-doing this figure completely;

- what is the purpose of plotting pupil size variability? From the methods, it looks that you used 8mm diameter black sticker as a reference for each participant, did you take into account the baseline pupil diameter for each individual? 

- Tables 2 and 3: make statistics tables easier to read (can put original version in the appendix and simpler in the text), how were the parameters of dependent vs pupil size measures compared?

- Make figure legends more self-explanatory rather than referring to the text

- Explain more in the discussion why is it useful to know that pupil size changes? How can it actually be applied in clinical practice?

Reviewer 3 Report

Van Egroo et al present a paper on the use of pupillometry (steady-state pupil size/pupil size variability) as a measure of sleepiness over a 29h of wakefulness. 20 participants underwent a constant routine procedure during which pupil size was measured in 2-minute recordings at regular intervals. Subjective measures of sleepiness, fatigue and mood were also collected, as well as objective vigilance performance, and resting EEG theta power. Results showed circadian modulation of subjective and objective measures of alertness, and of steady-state pupil size. Pupil variability, on the other hand, was not significantly altered by circadian phase/time awake. Furthermore, the authors report several correlational findings showing associations between subjective fatigue and pupil variability (interacting with circadian phase), and steady-state pupil size and subjective mood (motivation, sociability and anguish). No correlations with subjective sleepiness or vigilance performance were found.

Overall I think the study is well-conducted. It has several clear strengths, such as the highly controlled lighting environment and constant routine procedure, that help to analyze pupil metrics in relation to circadian factors and prolonged wakefulness. As such I think, it would provide a useful contribution to the literature.

I do think that the study has some limitations that could be addressed by further discussion and/or additional analysis.

1. First of all, I think the introduction/discussion of relevant literature is unnecessarily short (I’m not entirely sure, but I’m not aware of a specific word/reference limit for this paper). In particular, there is a substantial history of research into the relation between pupil size and sleep/arousal. Pioneering work e.g. from Lowenstein & Loewenfeld (1964), Yoss et al. (1970), work on ‘pupillary unrest index’ by Barbara and Helmut Wilhelm (1998a,b), and more recent work e.g. by (Maccora et al. 2018; Daguet et al. 2019), have discussed many aspects of pupil size in relation to sleep/extended wakefulness/circadian variation. It would be appreciated if a more complete and more detailed discussion of this literature would be provided to place the current study in the context of what is already known. At the moment this is limited to a very brief statement to the effect of: some studies show a relation (refs), but others don’t (refs).

2. Relatedly, I think there should be some distinction between spontaneous aspects of pupil size (e.g. steady-state size, pupil variability) as reflection of circadian/sleep modulation, and pupil responses that are probed by external events (e.g. pupil-light reflex). Mechanisms of modulation are not necessarily the same for both types of pupil behavior. Note that the current study includes spontaneous pupil metrics only.

3. As for the results, the one thing that stood out most for me was that the circadian pattern that was identified for pupil size seemed (i.e. sinusoidal pattern with a peak at 0 degrees and dips at -75 and 165-195 degrees circadian phase) to be different from the patterns  found for other metrics (i.e. mostly a monotonical increase in PVT RT and EEG theta; or single minimum/maximum around circadian phase 135-165 degrees for subjective ratings). This might suggest that pupil size is differently modulated by the combined effects of circadian and homeostatic drives than the other alertness measures. What would the authors make of these different circadian patterns? Would it be worth discussing in some more detail what maybe underlying this?

4. In the discussion the authors indicate that the short duration of the pupil recording (2 min) may have limited the sensitivity of the test to circadian fluctuation. I think this might be particularly true for pupil variability. Studies using the pupil unrest index, usually measure for longer durations (11 min, e.g. Wilhelm et al. 1998a,b). In our own research we find that pupil variability increases over the duration of a 10-min recording (Massar et al. 2019, cited in orig submission as Massar 2018). Could the authors comment on this?

5. For the PVT analysis, removing lapse trials is not the standard analysis practice, I believe. If this quantification is deemed more desirable by the authors, it might be useful to also report an analysis of the number of lapses. Related to this, I think the 500ms lapse criterion may be a bit too lenient for the auditory version of the PVT (e.g see Jung et al. 2011).

6. For the correlational analysis, it wasn’t fully clear to me whether between-subjects correlations or with-subjects correlations were modeled here (from fig 4b,c it seems it’s between-subjects). If it is between-subjects, I think a general note of caution, in that N = 20 is a rather small sample size to reliably assess associations between physiology and behavior.

7. I was a bit confused about some statements in the discussion.

P8 lines 208-210: “SCN driven changes in body temperature, via the subparaventricular zone and preoptic area [33], may also affect global brain activity [37], including at the level of the eye.” 

Do the authors mean that some circadian variation in pupil size could actually be driven by changes in body temperature? I couldn’t find any evidence of that in the two referenced papers.

P8 lines 210-215: “Sleep-wake history also affects global brain state notably through changes in the neurochemical environment of brain cells: increase in extracellular concentration of compounds such as adenosine, different ions and glutamate have been directly related to increased sleep need [38–40]. The latter chemical changes could also be present locally in the retina and drive local changes in pupil size. Evidence of local retinal mechanisms related to sleep-wake history driving retinal physiology remains scarce or inexistent, however.”

Regarding the last statement, is evidence ‘scarce’ (in that case please cite and discuss), or ‘inexistent’ (in that case, is there any credibility to this proposed mechanism)?

Daguet, I., Bouhassira, D., & Gronfier, C. (2019). Baseline Pupil Diameter Is Not a Reliable Biomarker of Subjective Sleepiness. Frontiers in Neurology, 10, 42–11. 

Jung et al. (2011). Comparison of sustained attention assessed by auditory and visual psychomotor vigilance tasks prior to and during sleep

deprivation. Journal of Sleep Research,20, 348-355.

Lowenstein, O., & Loewenfeld, I. E. (1964). The sleep-waking cycle and pupillary activity. Annals of the New York Academy of Sciences, 117(1), 142–156. 

Maccora, J., Manousakis, J. E., & Anderson, C. (2018). Pupillary instability as an accurate, objective marker of alertness failure and performance impairment. Journal of Sleep Research, 23(2), e12739. 

Massar et al. (2019). Sleep deprivation increases the cost of attentional effort: performance, preference, and pupil size. Neuropsychologia, 173, 169-177.

Wilhelm, B., Wilhelm, H., Lüdtke, H., Streicher, P., & Adler, M. (1998). Pupillographic assessment of sleepiness in sleep-deprived healthy subjects. Sleep, 21(3), 258–265.

Wilhelm, H., Lüdtke, H., & Wilhelm, B. (1998). Pupillographic sleepiness testing in hypersomniacs and normals. Graefe's Archive for Clinical and Experimental Ophthalmology, 236(10), 725–729. 

Yoss, R. E., Moyer, N. J., & Hollenhorst, R. W. (1970). Pupil size and spontaneous pupillary waves associated with alertness, drowsiness, and sleep. Neurology, 20(6), 545–554.

Author Response

Round  2

Reviewer 2 Report

All comments/ suggestions raised have been addressed by the authors.